

# A descriptive study of random forest algorithm for predicting COVID-19 patients outcome

Jie Wang[1], Heping Yu[2], Qingquan Hua[1], Shuili Jing[1], Zhifen Liu[3], Xiang Peng[4], Cheng'an Cao[4] and Yongwen Luo[5]

[1] Department of Otolaryngology-Head and Neck Surgery, Renmin Hospital of Wuhan University, Wuhan, Hubei, China
[2] Department of Nail and Breast Surgery, Wuhan Forth Hospital, Wuhan, Hubei, China
[3] Department of Nephrology, Wuhan Forth Hospital, Wuhan, Hubei, China
[4] Department of Neurosurgery, Wuhan Forth Hospital, Wuhan, Hubei, China
[5] Department of Urology, Zhongnan Hospital of Wuhan University, Wuhan, Hubei, China

Corresponding authors
Cheng'an Cao, cca24@163.com
Yongwen Luo, luoywen@whu.edu.cn

## ABSTRACT

**Background:** The outbreak of coronavirus disease 2019 (COVID-19) that occurred in Wuhan, China, has become a global public health threat. It is necessary to identify indicators that can be used as optimal predictors for clinical outcomes of COVID-19 patients.

**Methods:** The clinical information from 126 patients diagnosed with COVID-19 were collected from Wuhan Fourth Hospital. Specific clinical characteristics, laboratory findings, treatments and clinical outcomes were analyzed from patients hospitalized for treatment from 1 February to 15 March 2020, and subsequently died or were discharged. A random forest (RF) algorithm was used to predict the prognoses of COVID-19 patients and identify the optimal diagnostic predictors for patients' clinical prognoses.

**Results:** Seven of the 126 patients were excluded for losing endpoints, 103 of the remaining 119 patients were discharged (alive) and 16 died in the hospital.
A synthetic minority over-sampling technique (SMOTE) was used to correct the imbalanced distribution of clinical patients. Recursive feature elimination (RFE) was used to select the optimal subset for analysis. Eleven clinical parameters, Myo, CD8, age, LDH, LMR, CD45, Th/Ts, dyspnea, NLR, D-Dimer and CK were chosen with AUC approximately 0.9905. The RF algorithm was built to predict the prognoses of COVID-19 patients based on the best subset, and the area under the ROC curve (AUC) of the test data was 100%. Moreover, two optimal clinical risk predictors, lactate dehydrogenase (LDH) and Myoglobin (Myo), were selected based on the Gini index. The univariable logistic analysis revealed a substantial increase in the risk for in-hospital mortality when Myo was higher than 80 ng/ml (OR = 7.54, 95% CI [3.42–16.63]) and LDH was higher than 500 U/L (OR = 4.90, 95% CI [2.13–11.25]).

**Conclusion:** We applied an RF algorithm to predict the mortality of COVID-19 patients with high accuracy and identified LDH higher than 500 U/L and Myo higher than 80 ng/ml to be potential risk factors for the prognoses of COVID-19 patients in the early stage of the disease.

# INTRODUCTION

In December 2019, an outbreak of acute respiratory syndrome coronavirus (CoV) pneumonia occurred in Wuhan, Hubei Province, China (*Phelan, Katz & Gostin, 2020*), and attracted intense attention worldwide. The World Health Organization (WHO) named the virus, coronavirus disease 2019 (COVID-19), based on its identification from a patient's pharyngeal swab sample (*Lei et al., 2020*; *World Health Organization, 2020*). SAS-nCov 2 is a species of CoV, which is a family of the largest, enveloped, single-stranded, positive-sense RNA viruses (*Su et al., 2016*). The scientific community and infection control agencies face enormous challenges in controlling the increasing intensity of the COVID-19 pandemic. However, the disease has spread rapidly around the world. By 26 June 2020, COVID-19 had affected 213 countries, with over 9,621,470 confirmed cases and 487,295 deaths worldwide (COVID-19 CoV—Update https://virusncov.com/, accessed 26 June 2020). The least absolute shrinkage and selection operator (LASSO) regression has been used to identify the important factors of severity transition in COVID-19 patients (*Liu et al., 2020*), and critically ill patients exhibited respiratory failure, acute respiratory distress syndrome, heart failure, and septic shock, which increased the mortality of COVID-19 patients (*Zhou et al., 2020*).

Previous studies showed that patients who were elderly and had diabetes, cardiovascular disease, chronic respiratory diseases, or cancer presented an increased risk for COVID-19-related mortality worldwide (*Guan et al., 2020*; *Huang et al., 2020b*; *Ji et al., 2020*; *Wang et al., 2020*; *Wu & McGoogan, 2020*; *Zhou et al., 2020*). However, few models have been used to predict the mortality of COVID-19. Therefore, an effective and robust model is urgently required to predict the mortality of COVID-19 based on routine laboratory assessments and demographic information from COVID-19 patients. Timely detection of patients with high risk is of great significance and may contribute to optimizing the use of limited resources and delivering proper care.

Machine learning is widely used in medical diagnosis, and feature selection is an integral part of accurate data processing (*Guyon & Elisseeff, 2003*). Random forest (RF) is a type of machine learning that can analyze complex interactions between clinical characteristics and provide high classification accuracy using a set of decision trees (*Touw et al., 2013*). Therefore, we used a risk RF prediction model based on the outcomes of COVID-19 patients to predict the likelihood of recovery or continued deterioration and speculated on their prognoses, and corresponding disease control strategies need to be stressed to protect these patients against SAS-nCov 2.

# PATIENTS AND METHODS

## Study design and participants

This was a retrospective cohort analysis that included 126 patients, aged 27–87 years, from Wuhan Fourth Hospital. These patients were diagnosed with COVID-19 based on the World Health Organization's interim guidelines. Of the 126 patients, seven patients were

excluded due to a lack of known clinical endpoints. The remaining 119 patients in this study were hospitalized for treatment from 1 February to 15 March 2020. This study was approved by the Ethics Committee of Wuhan Fourth Hospital (KY 2020-032-01). The Hospital Ethics Committee waived the informed consent from the study participants due to the high transmissibility of the disease.

Among the patients, the criteria used for discharge were as follows. The highest temperature returned to normal for more than three days. Chest CT imaging revealed significant inflammation absorption, and the respiratory symptoms had substantially improved. Two consecutive nucleic acid tests from throat swabs were negative, and the time interval between testing was at least one day. Finally, after evaluation and a unanimous decision by the expert team, a patient was discharged.

## Data collection

Clinical information for all patients was obtained from electronic medical records in Wuhan Fourth Hospital by three independent researchers. Patient information, including exposure history, demographics, medical history, laboratory findings, co-morbidities and clinical outcomes were collected and analyzed. If data were missing from the medical records, we obtained data from attending doctors or directly from the patients. Access was granted by the director of the hospital.

## Statistical analysis

Descriptive data were compared using quartiles and medians, and the $\chi^2$ test or Fisher's exact test was used to analyze categorical data. The Kolmogorov–Smirnov test was used to analyze the normality of the data from discharged patients ($n = 103$). The Shapiro–Wilk test was employed to analyze the normality of the data from patients who died ($n = 16$). Subsequently, normally distributed laboratory results were analyzed using independent sample paired $t$-tests. The nonparametric Mann–Whitney–Wilcoxon test was used for data that did not exhibit a normal distribution. Univariable logistic analysis was used to analyze the risk of mortality caused by two variables in survival and non-survival patients. All data were assessed using IBM SPSS, Version 26.0.

## Variable selection and model construction

The flowchart for the research design is shown in Fig. S1. Based on the imbalanced distribution of COVID-19 discharged versus deceased patients (106:13), a SMOTE procedure was employed to adjust the data to achieve a final ratio of 1:1 (103:103). Spearman correlation was used to calculate the correlations among the essential variables, which were chosen based on statistical analysis. RFE was used to screen out the discriminative subset of COVID-19 patient clinical characteristics with 10-fold cross-validation to avoid the redundant information. Then an RF classification model was used to predict the mortality of COVID-19 patients with 5-fold cross-validation and the mean value of the accuracy of mtry = 1 was the highest at 0.846, a bagging algorithm was used to randomly collect the clinical characteristic for a total of 500 times, the Gini index was the split criterion, and the nodesize of RF classification model was 1. The clinical data were divided

into a training set and a test set with a ratio of approximately 4:1 (166:40). All data were processed using R studio (R 3.6.3), the entire workflow was processed with the caret package (http://CRAN.R-project.org/package=caret) to keep the model construction and validation consistent.

## Correlation analysis and assessment of accuracy

A partial dependence correlation analysis was employed to provide a graphical depiction of the marginal effect of a variable on the COVID-19 patients' outcome during the calculation process (*Greenwell, 2017*). The function being plotted was defined as:

$$\tilde{f}_{(x)} = \frac{1}{n}\sum_{n=1}^{n} f(x, x_{ic})$$

where $x$ is the variable corresponding to the chosen clinical characteristic, and $x_{ic}$ represents the other variables in the clinical information. The summand was the predicted logits (log of a fraction of votes) for classification:

$$f(x) = \log_{Pk}(x) - \frac{1}{k}\sum_{j=1}^{k} \log_{Pj}(x)$$

where $K$ is the number of classes, and $P_j$ is the proportion of votes for class $j$.

The accuracy of the test group to identify the final diagnostic capability of the RF classification algorithm was assessed using AUC, it also was applied to choose the optimal mtry and the best subset of clinical characteristics for the RF model performance.

# RESULTS

## Clinical demographics and outcomes of COVID-19 patients

We described a cohort of 126 patients hospitalized at the Wuhan Fourth Hospital between 1 February to 15 March 2020, of whom approximately half were classified as severely ill or critically ill. The patient clinical demographics and outcomes are shown in Table 1. We found that 48 patients (38.1%) were older than 65, and the median patient age was 60 years (IQR 53-69.5), which showed the prevalence of COVID-19 in older adults. The incidence of COVID-19 infection was gender-neutral in that the proportions of male and female patients were nearly identical. COVID-19 patients typically exhibited fever (92.0%) and 39 (34.8%) patients had peak temperatures above 39 °C. The most frequently observed symptoms of COVID-19 patients on admission were cough (75.4%), followed by fatigue (58.7%), dyspnea (55.6%). In addition, many patients suffered from co-morbidities with hypertension (34.9%) being the most common co-morbidity, followed by diabetes (16.7%), cardiovascular and macrovascular disease (11.9%). During treatment, 83 patients (65.9%) used nasal cannulas for supplemental oxygen, indicating that a nasal cannula was useful for COVID-19 patients, two additional respiratory support strategies that were used were noninvasive mechanical ventilation (NMV) (27.8%), and invasive mechanical ventilation (IMV) (4.0%). The criteria used to determine illness severity were based on the Novel CoV Pneumonia Prevention and Control Program

**Table 1 Demographic characteristics and clinical outcomes of patients with COVID-19.**

| Variable | Number of patients (%) |
|---|---|
| No. of patients | 126 |
| Age, median (IQR), y | 60 (53–69.5) |
| ≥65 | 48 (38.1) |
| <65 | 78 (61.9) |
| Highest patient temperature, median (IQR), °C | 38.6 (38–39) |
| ≥39 (high fever) | 39 (34.8) |
| <39 | 73 (65.2) |
| Gender | |
| Male | 65 (51.6) |
| Female | 61 (48.4) |
| Contact history of epidiemic area | 7 (5.6) |
| Initial common symptoms | |
| Fever | 112 (88.9) |
| Cough | 95 (75.4) |
| Productive cough | 21 (16.7) |
| Hemoptysis | 6 (4.8) |
| Dyspnea | 70 (55.6) |
| Fatigue | 74 (58.7) |
| Myalgia | 41 (32.5) |
| Diarrhea | 14 (11.1) |
| Comorbidities | |
| Hypertension | 44 (34.9) |
| Diabetes | 21 (16.7) |
| Cardriovascular and Macrovascular disease | 15 (11.9) |
| Liver and gall disease | 5 (4.0) |
| Nervous system disease | 6 (4.8) |
| Chronic lung disease | 13 (10.3) |
| Chronic kidney disease | 3 (2.4) |
| Endocrine system disease | 2 (1.6) |
| Immunological disease | 1 (0.8) |
| Hyperlipidemia | 3 (2.4) |
| Gastric disease | 7 (5.6) |
| Tumor | 6 (4.8) |
| Highest level of oxygen therapy | |
| Nasal cannula | 83 (65.9) |
| NMV | 35 (27.8) |
| IMV | 5(4.0) |
| IMV with ECMO | 0 |
| Severity of clinical condnition | |
| Moderate | 61 (50.0) |
| Severe | 38 (31.1) |
| Critical | 23 (18.9) |

(Continued)

| Table 1 (continued) | |
|---|---|
| **Variable** | **Number of patients (%)** |
| Clinical outcomes | |
| Discharged alive | 103 (86.6) |
| Died | 16 (13.4 ) |

**Note:**

IQR, interquartile range; NMV, noninvasive mechanical ventilation (including high flow supply and face mask); IMV, invasive mechanical ventilation; ECMO, extracorporeal membrane oxygenation.

(sixth edition) published by the National Health Commission of China (http://www.gov.cn/zhengce/zhengceku/2020-03/04/5486705/files/ae61004f930d47598711a0d4cbf874a9.pdf). Among the patients included in this study, in addition to seven patients who were excluded due to lack of known clinical endpoints, 103 (86.6%) patients were cured, and 16 (13.4%) patients died.

## Laboratory findings of COVID-19 patients at admission

The initial laboratory findings included complete blood count, serum biochemical tests, coagulation profiles, and myocardial enzymes. All patients were assessed to determine whether they deviated significantly ($p < 0.05$) from a normal range to evaluate the status of important organ functions. As seen in Table 2, more than 80% of the patients exhibited lymphopenia, especially for CD4+ and CD8+ T lymphocytes (91.3%), which confirmed the previous study that SARS-CoV-2 infection damaged the immune system (*Huang et al., 2020a*). Approximately half of the patients exhibited decreased Th/Ts ratios, which differed from Middle East respiratory syndrome(*Park et al., 2017*). C-reactive protein (CRP) was elevated in 85.6% of the patients, and procalcitonin (PCT) was slightly increased. COVID-19 infection also impaired coagulation functions in some patients. In this study, the prothrombin time was prolonged in approximately half of the patients, fibrinogen (FIB) was increased in two-thirds of the patients, and D-dimer was increased in 76.2% of the patients. The severely infected COVID-19 patients displayed a trend towards reduced platelet counts, a higher D-dimer level, and a higher rate of DIC occurrence. The myocardial enzymes showed that myocardial cell injury occurred in some patients, as 60% exhibited elevated B-type natriuretic peptide (BNP) and patients with elevated LDH accounted for 76.2% of the total.

## Comparison of clinical characteristics between discharged and deceased patients

A comparison of clinical characteristics between discharged (alive) and deceased patients revealed the significant features that most likely caused deterioration in the deceased patients. As seen in Table 3, the patients in the deceased group were older than those in the discharged group ($p < 0.001$), and the majority were males (75%). The proportion of patients who experienced dyspnea was apparently increased from the onset of illness in the deceased group ($p = 0.018$). The deceased patients were more susceptible to respiratory failure, and the arterial blood gas parameters, PCO2, PO2, SO2, and oxygenation at admission, were significantly reduced in the deceased group. The laboratory analysis in

**Table 2** Initial laboratory indices of patients infected with COVID-19.

| Laboratory Indices | Reference values | Number of all patients | Median (IQR) | Number of patient with value deviation from reference (%) |
|---|---|---|---|---|
| Hematology | | | | |
| White blood cells, $\times 10^9$/mL | 3.5–9.5 | 126 | 6.14 (3.96–8.29) | 26 (20.6)[a] |
| Neutrophils, $\times 10^9$/mL | 1.8–6.3 | 126 | 4.51 (2.77–7.34) | 41 (32.5)[a] |
| Lymphocytes, $\times 10^9$/mL | 1.1–3.2 | 126 | 0.73 (0.53–1.01) | 102 (81.0)[b] |
| Monocytes, $\times 10^9$/m L | 0.1–0.6 | 126 | 0.29 (0.20–0.43) | 6 (4.8)[a] |
| NLR | NA | 126 | 5.99 (3.07–12.59) | |
| LMR | NA | 126 | 2.39 (1.65–3.68) | |
| CD4+ Tlym, $\times 10^6$/mL | 450–1,440 | 116 | 142.16 (78.50–271.84) | 108 (93.1)[b] |
| CD8+ Tlym, $\times 10^6$/mL | 320–1250 | 116 | 109.84 (61.35–154.52) | 108 (93.1)[b] |
| Th/Ts | 1.5–2.9 | 116 | 1.52 (0.96–2.08) | 55 (51.9)[b] |
| CD45, $\times 10^6$/mL | NA | 116 | 481.92 (338.36–724.95) | |
| Biochemical analysis | | | | |
| AST, U/L | 15–40 | 126 | 27.5 (19–44) | 48 (38.1)[a] |
| ALT, U/L | 9–50 | 126 | 25 (15–43.5) | 22 (17.5)[b] |
| TG, mmol/L | 0.45–1.69 | 126 | 1.49 (1.16–1.89) | 41 (32.5)[a] |
| Creatine, μM | 57–111 | 126 | 66 (54–81.25) | 7 (5.6)[a] |
| TnI, μg/L | 0–0.6 | 91 | 0.03 (0.03–0.03) | 1 (1.1)[a] |
| Myo, ng/mL | 0–80 | 111 | 27.2 (18.1–38.05) | 13 (11.7)[a] |
| CK, U/L | 0–171 | 119 | 63.2 (35.25–138.05) | 18 (15.1)[a] |
| CK-MB, ng/m L | 0–2.37 | 92 | 1.1 (1–2.33) | 23 (25.0)[a] |
| BNP, ng/mL | 0–100 | 88 | 196.5 (42.25–754.25) | 53 (60.2)[a] |
| CEA , μg/L | 0–5 | 57 | 2.08 (1.51–5.53) | 15 (26.3)[a] |
| LDH, U/L | 120–150 | 126 | 306.50 (241–389) | 123 (97.6)[a] |
| Infection indices | | | | |
| CRP, mg/L | 0–5 | 126 | 40.31 (21.27–86.56) | 166 (85.6)[a] |
| PCT, ng/mL | 0–0.5 | 121 | 0.04 (0.04–0.08) | 5 (4.1)[a] |
| Coagulation function | | | | |
| PT, s | 9–13 | 126 | 13.6 (11.3–41.2) | 61 (48.4)[a] |
| APTT, s | 20–40 | 126 | 35.5 (22.4–69.9) | 22 (17.4)[a] |
| TT, s | 14–21 | 126 | 16.3 (12.8–72.3) | 3 (2.4)[a] |
| INR | 0.8–1.25 | 126 | 1.10 (0.86–4.33) | 7 (5.6)[a] |
| FIB, g/L | 2–4 | 126 | 4.79 (1.01–37.9) | 83 (65.9)[a] |
| D-Dimer, mg/L | 0–0.2 | 126 | 1.96 (0.03–60.14) | 96 (76.2)[a] |

Notes:
[a] Above reference.
[b] Below reference.
IQR, interquartile range; NLR, neutrophil lymphocyte ratio; LMR, lymphocyte monocyte ratio.

this study revealed that more patients in deceased group exhibited elevated neutrophils and lymphopenia, which indicated a "divergence" between these two variables. Therefore, the neutrophil to lymphocyte ratio (NLR) was used to indicate the severity of illness in the patients. The NLR was remarkably elevated in the deceased group, while the

Table 3 Comparison of clinical characteristics between the discharged and deceased groups.

| Variables | No. of patients | | Deceased (n = 16) | Statistics | p-Value |
|---|---|---|---|---|---|
| Demographics | | | | | |
| Male | 61 | 49 (52.8) | 12 (8.2) | 4.170[d] | 0.041 |
| Female | 58 | 54 (50.2) | 4 (7.8) | | |
| Age | 119 | 58.65 ± 1.21 | 71.81 ± 1.85 | −5.948[a] | 0.000 |
| Peak temperature | 106 | 38.54 ± 0.06 | 38.73 ± 0.16 | 1.645[b] | 0.100 |
| Dyspnea | | | | | |
| Yes | 68 | 54 (58.9) | 14 (9.1) | 5.598[d] | 0.018 |
| No | 51 | 49 (44.1) | 2 (6.9) | | |
| Fatigue | | | | | |
| Yes | 72 | 61 (62.3) | 11 (9.7) | 0.526[c] | 0.468 |
| No | 47 | 42(40.7) | 5(6.3) | | |
| Hematology | | | | | |
| WBC, ×10$^9$/mL | 119 | 6.49 ± 0.37 | 8.22 ± 1.22 | 1.079[b] | 0.281 |
| Neu, ×10$^9$/mL | 119 | 5.21 ± 0.34 | 7.24 ± 1.18 | 2.060[a] | 0.042 |
| Lym, ×10$^9$/mL | 119 | 0.87 ± 0.06 | 0.58 ± 0.07 | −2.551[b] | 0.11 |
| Mon, ×10$^9$/mL | 118 | 0.32 ± 0.02 | 0.35 ± 0.04 | 1.091[a] | 0.275 |
| NLR | 119 | 8.46 ± 0.83 | 16.16 ± 3.15 | −2.785[b] | 0.005 |
| LMR | 119 | 3.29 ± 0.22 | 1.64 ± 0.27 | 2.880[a] | 0.005 |
| Total Tlym, ×10$^6$/mL | 109 | 369.89 ± 27.62 | 168.71 ± 27.53 | −3.677[b] | 0.000 |
| CD4+ Tlym, × 10$^6$/mL | 109 | 202.67 ± 15.38 | 115.62 ± 22.70 | −2.741[b] | 0.006 |
| CD8+ Tlym, × 10$^6$/mL | 109 | 150.17 ± 12.37 | 51.35 ± 7.92 | 4.468[b] | 0.000 |
| Th/Ts | 109 | 1.57 ± 0.10 | 2.45 ± 0.32 | 2.951[b] | 0.003 |
| CD45, × 10$^6$/mL | 109 | 635.82 ± 43.43 | 346.70 ± 57.66 | 3.070[b] | 0.002 |
| Biochemical analysis | | | | | |
| AST, U/L | 119 | 32.3 ± 1.8 | 41.9 ± 5.7 | −1.831[b] | 0.067 |
| ALT, U/L | 119 | 33.6 ± 3.0 | 42.1 ± 9.1 | −1.266[b] | 0.205 |
| TG, mmol/L | 119 | 1.64 ± 0.07 | 1.57 ± 0.13 | 0.055[b] | 0.957 |
| Creatine, μM | 119 | 69.76 ± 2.91 | 82.25 ± 0.88 | −1.611[a] | 0.110 |
| Myo, ng/mL | 111 | 31.80 ± 3.19 | 109.4 ± 23.93 | −10.77[b] | 0.000 |
| CK, U/L | 112 | 100.33 ± 14.21 | 152.73 ± 30.36 | −2.354[b] | 0.019 |
| CK-MB, ng/mL | 85 | 2.37 ± 0.44 | 2.89 ± 0.58 | −2.250[b] | 0.024 |
| Infection indices | | | | | |
| LDH, U/L | 119 | 312.95 ± 12.54 | 481.94 ± 43.23 | −3.981[b] | 0.000 |
| CRP, mg/L | 119 | 49.49 ± 3.91 | 67.37 ± 10.38 | −1.753[a] | 0.080 |
| Coagulation function | | | | | |
| PCT, ng/mL | 114 | 0.07 ± 0.10 | 0.20 ± 0.24 | −2.610[b] | 0.009 |
| APTT, s | 119 | 35.06 ± 0.62 | 36.80 ± 0.06 | 1.176[b] | 0.239 |
| TT, s | 118 | 15.99 ± 0.27 | 15.72 ± 0.43 | −0.830[b] | 0.407 |
| PT, s | 119 | 13.58 ± 0.31 | 14.19 ± 0.66 | −1.068[b] | 0.286 |
| INR | 119 | 1.09 ± 0.03 | 1.13 ± 0.05 | −1.169[b] | 0.242 |
| FIB, g/L | 119 | 4.82 ± 0.85 | 4.64 ± 0.40 | 0.651[b] | 0.515 |
| D-Dimer, mg/L | 115 | 1.25 ± 0.29 | 3.19 ± 1.27 | −3.003[b] | 0.003 |

| Variables | No. of patients | | | Deceased (n = 16) | Statistics | p-Value |
|---|---|---|---|---|---|---|
| Blood gas analysis | | | | | | |
| PH | 119 | | 7.43 ± 0.01 | 7.41 ± 0.04 | 0.507[a] | 0.619 |
| PCO2, mmHg | 118 | | 38.75 ± 0.47 | 33.80 ± 1.91 | 2.514[a] | 0.023 |
| PO2, mmHg | 119 | | 81.98 ± 3.07 | 55.06 ± 3.49 | −3.837[b] | 0.000 |
| SO2, % | 119 | | 93.88 ± 0.47 | 83.63 ± 4.83 | −3.582[b] | 0.029 |
| oxygenation, mmHg | 119 | | 282.8 ± 13.9 | 124.3 ± 10.6 | 9.072[a] | 0.000 |

Notes:
[a] $t$-test.
[b] Mann–Whitney $U$ test.
[c] $\chi^2$ test.
[d] Continuity Correction.
NLR, neutrophil lymphocyte ratio; LMR, lymphocyte monocyte ratio.

lymphocyte to monocyte ratio (LMR) was decreased in the deceased group. The immune system damage was a risk factor for unfavorable outcomes of the disease. T lymphocytes significantly decreased in the deceased group, especially CD4+ and CD8+, as did the Th/Ts ratio. Compared to discharged patients, the deceased patients underwent more frequent myocardial cell injury, as parameters reflecting heart function, including Myo, CK and LDH, were significantly increased in the deceased group. Moreover, the inflammation-related indices, CRP ($p = 0.080$) and PCT ($p = 0.009$) were significantly higher in the deceased group.

## Correlation between clinical characteristics

Because imbalanced data distribution affects the prediction accuracy of the RF model, and the ratio of discharged versus deceased patients was 103:16, a SMOTE algorithm was used to balance the data to select a more representative and informative subset of parameters for COVID-19 patients. STOME adjusted the ratio between these two groups to achieve a ratio of 1:1 (103:103). Moreover, redundancy of information also is likely to decrease the prediction performances of the RF classification model, so the correlations between variables should be taken into account in the process of feature selection (*Paul et al., 2017*). A Spearman correlation coefficient test was used to analyze the correlation between clinical characteristics of the COVID-19 patients (*Spearman, 2010*). A heatmap was used to show the correlations between variables in the form of a matrix in Fig. 1. Each element in the matrix was the correlation coefficient between the variables, and the range (−1, 1) was used to evaluate degree of correlation between two variables. When the correlation coefficient was greater than 0.8, and the $p$-value was smaller than 0.05, the correlation was determined to be strong (*Paul et al., 2017*), indicating that the factors were redundant variables. The analysis revealed that CD45 and CD4 had a high correlation of 0.84 ($p < 0.01$), meanwhile, NLR and neutrophils also had a correlation of 0.84 ($p < 0.01$). As for the redundancy of the clinical features, further processing was required to select the best subset for the RF model.
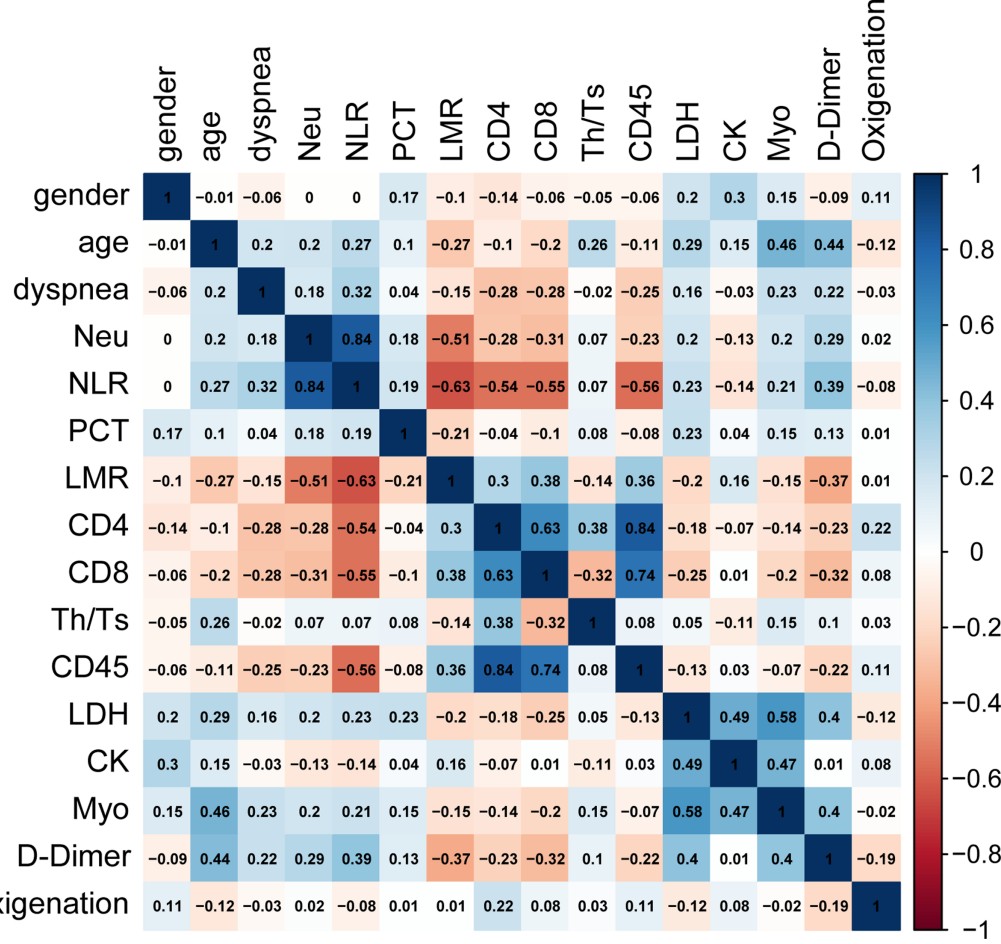

**Figure 1 Spearman correlation analysis between clinical features of COVID-19 patents.** Heat map visualization shows Spearman correlation coefficients for clinical characteristics, the variables have high correlations when ρ > 0.8 and $p < 0.05$. The analysis indicated that the variables, CD45 and CD4, NLR and Neu, were highly correlated.               

## Variable selection and RF classification model construction

To select the optimal subset of clinical features, an RFE processed with 10-fold cross-validation was used to select the best subset. The RFE could eliminate the redundant and irrelative information from the COVID-19 patients and enhance the performance of the RF classification model (*Darst, Malecki & Engelman, 2018*). The results selected 11 clinical characteristics, Myo, CD8, age, LDH, LMR, CD45, Th/Ts, dyspnea, NLR, D-Dimer and CK with the highest accuracy at 0.9905 (Fig. 2A), which revealed the optimal complexity of the feature subset. Next, the RF classification model was used to predict the prognoses of COVID-19 patients based on the best subset. Five-fold cross-validation was used to identify the optimal mtry for the RF classification model, and the highest accuracy of classification was mtry = 1 (Fig. 2B), with the highest corresponding mean value of AUC at 0.846. Moreover, out-of-bag (OOB) error represented the generalization ability of the RF to calculate the proportion of misclassification. In Fig. 2C, the OOB error gradually decreased and stabilized as the forest size increased, and it finally reduced to

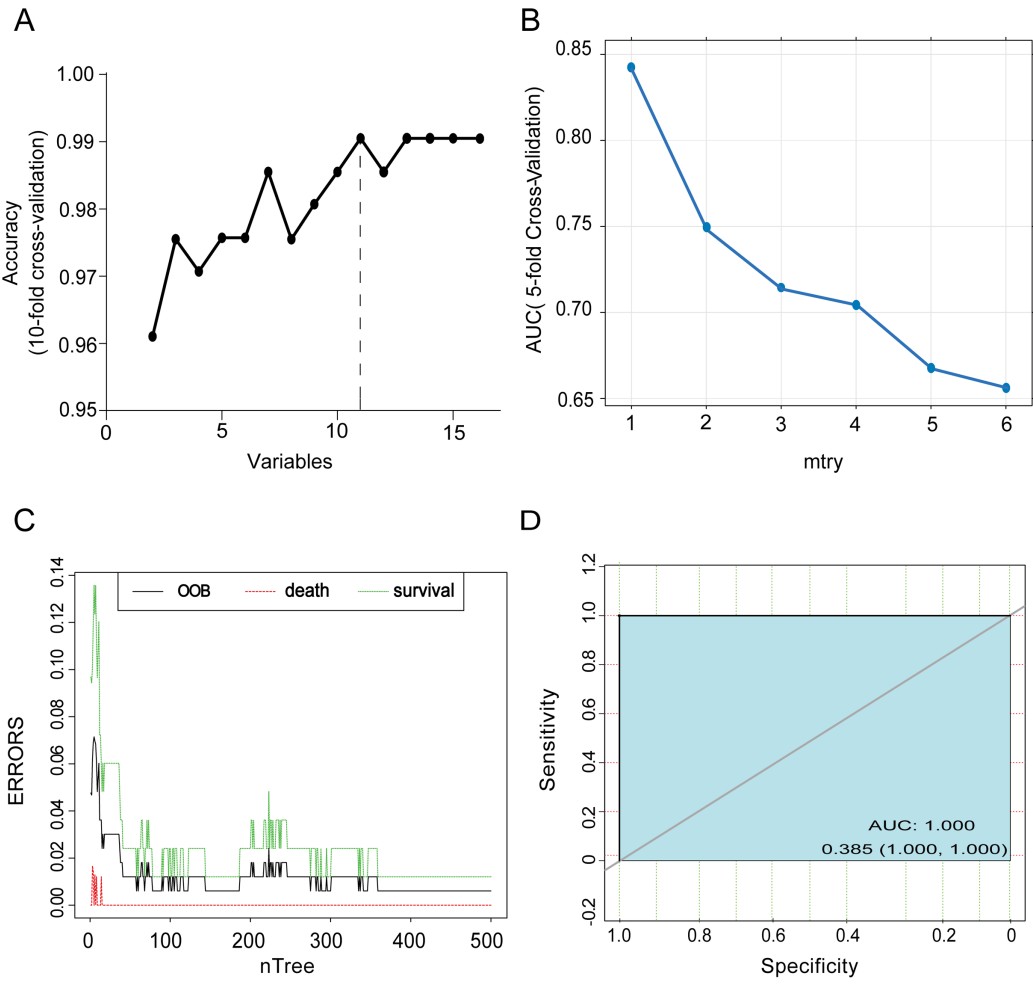

**Figure 2 Variable selection and accuracy tests for the RF classification model.** (A) The process of choosing the best subset of clinical features. Eleven variables were proved to be the optimal selection with the corresponding highest accuracy (0.9905) of a predictive result after 10-fold cross-validation. (B) Mtry = 1 was the best choice with the highest accuracy (0.846) for RF model prediction after 5-fold cross-validation. (C) The OOB error rate to evaluate the quality of RF prediction of COVID-19 patients' outcomes. The middle line depictes the OOB error rate of all data. The top line showes the OOB error in the subgroup of survival patients and the bottom line showes OOB error in the subgroup of dead patients. (D) ROC curve shows the accuracy of test data in the RF classification models, and the threshold of it is 0.385

less than 0.05 when the tree number reached 500. Meanwhile, death and survival errors gradually reduced to the same level as the OOB. The final diagnostic capability of the RF classification calculations was assessed using the test group's accuracy, which was 100% (Fig. 2D), the threshold for the test data ROC was 0.385.

## Identification of the important predictors for clinical outcomes

As we know, the result of the RF classification model was obtained by selecting the results of the combined predictions among 500 decision trees, and the Gini index was the split criterion. RF-Gini is one of the best methods for feature ranking worldwide, especially for the top five predicted features (*Menze et al., 2009*). The larger the Gini coefficient became,

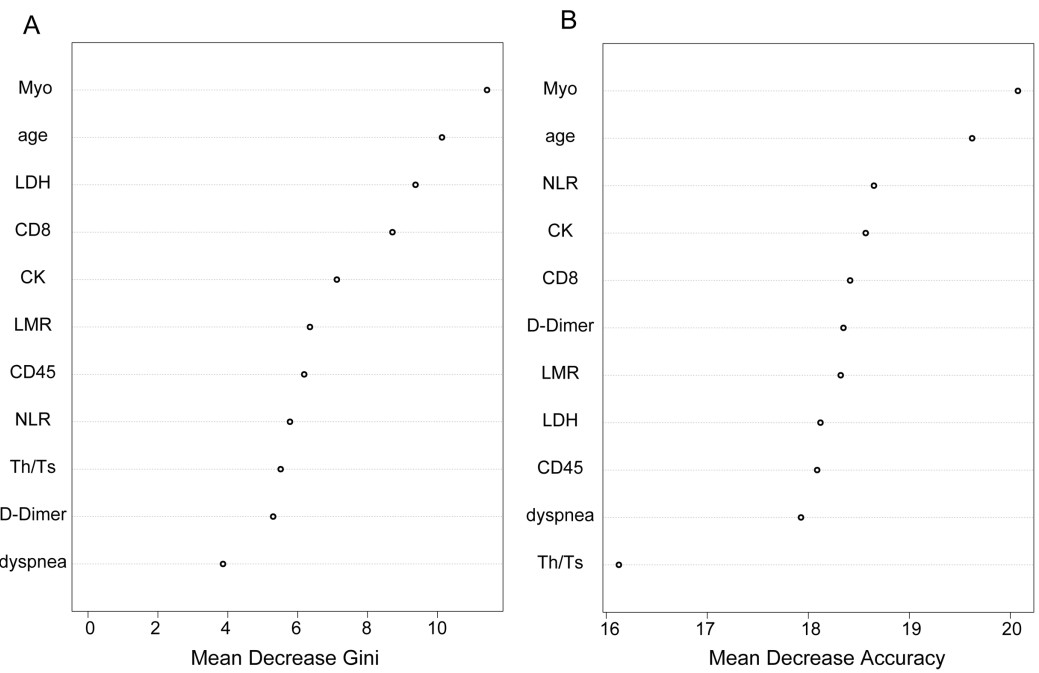

**Figure 3 Identification of optimal diagnostic clinical characteristics for the prognoses of COVID-19 patients.** (A) Ranking of the clinical characteristics, according to Gini index. (B) Ranking of the clinical characteristics based on the standardized drop in prediction accuracy.

the more important the information content of the independent variables. As shown in Fig. 3A, the variables that were ranked as important included Myo, age, LDH, CD8, CK, LMR, CD45, NLR, Th/Ts, D-dimer and dyspnea. The top five variables were Myo, age, LDH, CD8, CK, among them, we chose Myo and LDH as two laboratory parameters to assess risk and indicate the prognoses for COVID-19 patients. The accuracy of the variables screened by the RF model is shown in Fig. 3B, and the accuracy of Myo ranked the highest and was followed by age and NLR.

## Relationship between clinical characteristics and survival in COVID-19 patients

To further analyze the role of LDH and Myo in affecting the survival of COVID-19 patients, we compared the mortality of patients who exhibited different levels of LDH and Myo. Using univariate logistic analysis, a substantial increase in the risk of in-hospital mortality with increased levels of Myo and LDH was observed (Fig. 4A). Patients with increased Myo (≥80 ng/ml) exhibited a 7.54-fold (95% CI [3.42–16.63]) increase in mortality compared to patients with low Myo (< 80 ng/ml). Similarly, patients with increased LDH (≥500 U/L) exhibited a 4.90-fold (95% CI [2.13–11.25]) increase in mortality compared to patients with low LDH (<500 U/L). The levels of LDH and Myo were compared in discharged and deceased groups (Fig. 4B). The median and IRQ for these two variables in the deceased group were higher than in the discharged group ($p < 0.001$). The partial dependence plot showed the impact of Myo and LDH on survival

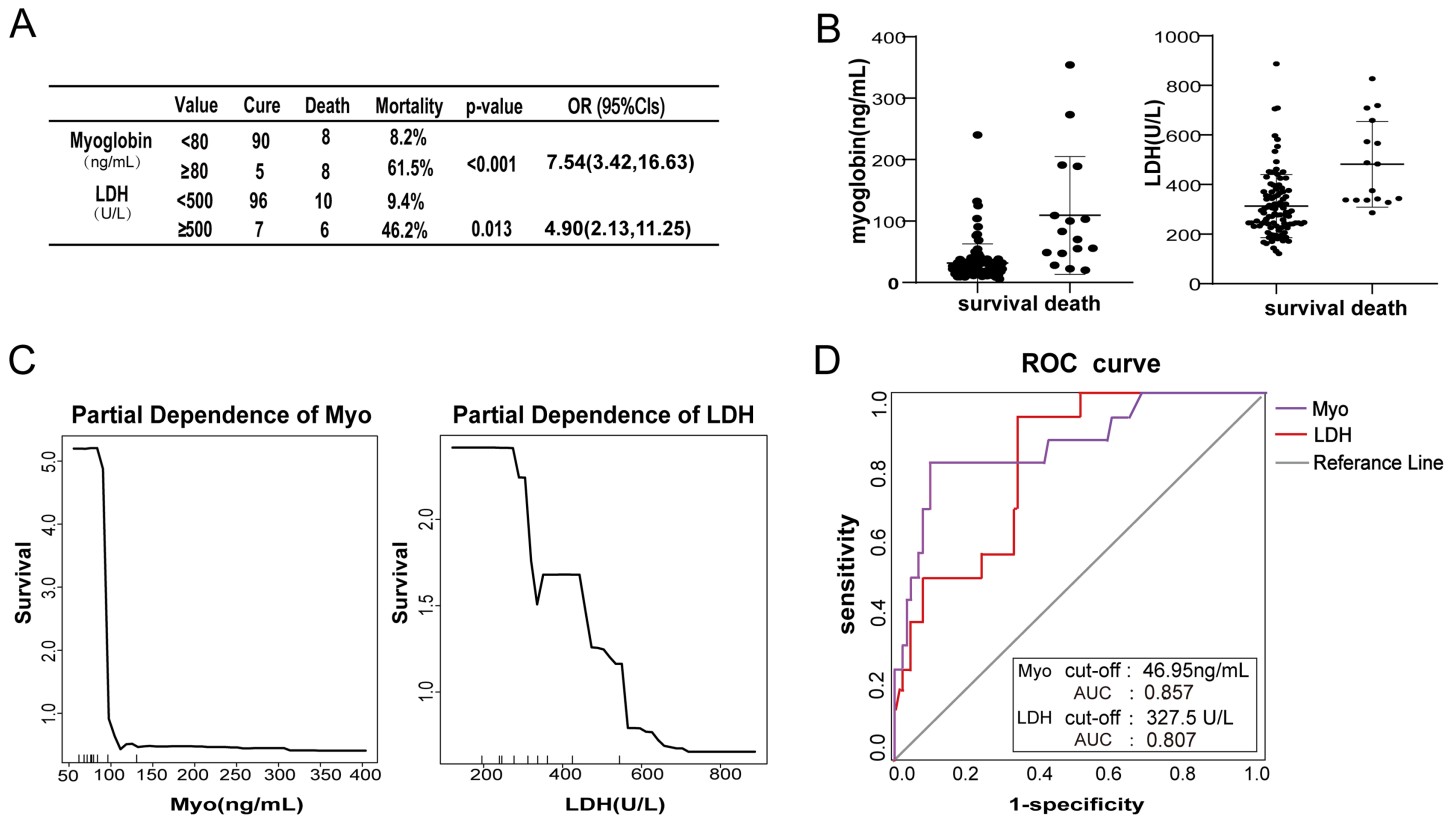

**Figure 4 Relationship between clinical characteristics and survival in COVID-19 patients.** (A) The table shows that the mortality rate increased significantly as the levels of Myo and LDH increased. (B) The scatter plot shows the different concentrations of Myo and LDH in the non-survival and survival groups. (C) The tendency chart shows the partial dependence correlation of Myo and LDH with survival. (D) ROC curve shows the accuracy of Myo and LDH in predicting the COVID-19 patients' outcome.

when the marginal effects were controlled for in the RF classification. As Fig. 4C shows, there was a significant negative correlation between survival and the levels of LDH or Myo. Specifically, increased levels of LDH and Myo were precursors to a poor prognosis for COVID-19 patients. To test the ability of LDH and Myo to predict the outcome of COVID-19 patients (Fig. 4D), we observed that the AUC for Myo was 0.857 (95% CI [0.7466–0.9694]), and the threshold for the Myo ROC curve was 46.95 ng/mL; the AUC for LDH was 0.807 (95% CI [0.7149–0.9053]), and the LDH ROC curve threshold was 327.5 U/L. These clinical features all had high accuracy for prognoses prediction of COVID-19 patients, but their accuracy was lower than that produced by the RF classification model.

## DISCUSSION

The COVID-19 virus that occurred in Wuhan, China, is highly contagious, and a large number of exposed people have become critically ill. This study provided a comprehensive description of the demographics, comorbidities, and laboratory findings, of COVID-19 patients. Among the laboratory results, it was observed that lymphocytes, including CD4+ and CD8+, were decreased in 91.3% of patients, which confirmed that SARS-CoV-2 infection injured the human immune system. Thus, subsequent immune responses to this

virus may exacerbate the disease response (*Huang et al., 2020a*). Approximately 40% of the COVID-19 cases were severe, and the disease resulted in a 13.4% mortality rate. The mortality rate observed in this study was higher than the average rate observed in Wuhan, which was 4% by 24 March 2020 (*Du et al., 2020*). This difference was likely due to the fact that only severe patients could be transferred to the hospitals designated to treat COVID-19 patients. During the patients' hospitalization for treatment, we found that early COVID-19 symptoms are insidious, but the disease progression is fast. Therefore, early prediction of COVID-19 patients' outcomes and adopting appropriate treatment are urgently required.

In this study, significant clinical features ($p < 0.05$) were identified between discharged and deceased patients using statistical analysis. These clinical features were used for RF classification model, with SMOTE and a feature reduction technique RFE, to predict mortality of COVID-19 patients. The AUC of the RF model reached 100%, demonstrating its robust prediction ability. Moreover, Myo and LDH were identified as two optimal predictors of COVID-19 patients' outcomes with the Gini index.

Our current studies suggested that the deceased patients were susceptible to multiple organ failure, especially heart and respiratory failure. There are several potential reasons for myocardial cell injury in COVID-19 patients, including systemic inflammatory responses, ACE2-targeted SAS-Cov-2 attacks on myocardial and lung cells, adverse effects of some anti-virus drugs (*Clerkin et al., 2020*), and some underlying myocardial-damaging co-morbidities, such as diabetes and hypertension. Previous studies have reported that heart injury was common in patients with pneumonia (*Marrie & Shariatzadeh, 2007*). It was reported that elevated concentrations of Myo in venous blood could predict the severity of COVID-19 (*McRae et al., 2020*). Myo is a significant myocardial marker that is used in the clinical detection of patients with severe pneumonia. In this study, we found that 75% of the non-survivors, whose Myo concentration was higher than 80 ng/mL, exhibited hypertension, which might have accelerated myocardial injury in patients with COVID-19. A partial correlation also indicated that as the Myo concentrations increased, survival decreased. Moreover, based on univariate logistic analysis, a high level of Myo above 80 ng/mL correlated with a high mortality rate (61.5%) from COVID-19, and the risk of mortality was increased by 7.54 (95% CI [3.42–16.63]), suggesting that increased concentrations of Myo potentially led to poor outcomes.

Lactate dehydrogenase is another indicator that reflects the degree of tissue damage caused by the virus and disease severity, including damage to myocardial (*Mamas, Fraser & Neyses, 2008*; *Warren-Gash, Smeeth & Hayward, 2009*), muscle, and lung cells. On the one hand, COVID-19 patients have severely reduced lung ventilation, which leads to hypoxia and carbon dioxide retention(*Yang et al., 2020b*), which damages tissues (*Yang et al., 2020a*). On the other hand, microcirculation disorders caused by the infection and insufficient tissue perfusion also result in tissue damage. Both processes lead to LDH accumulation in circulating blood. In our current research, the LDH values of 96 (76.2%) patients were higher than the normal reference range, and the average level of LDH for non-survivors was higher than survivors. LDH concentrations higher than 500 U/L were associated with high mortality risk (OR = 4.90, 95% CI [2.13–11.25]).
Moreover, in severely affected patients, abnormal elevation of LDH often indicates rapid disease progression and acute respiratory failure (*Zhang et al., 2020*). Therefore, an increase in LDH was a significant risk factor for COVID-19 patient mortality.

Given the above research, our findings suggest that strategies to protect organ function should be emphasized to improve the patients' survival. Except for ensuring that they remain protected from getting infected, doctors should evaluate the cardiac condition and the degree of complications of patients before and during treatment to avoid adverse side-effects from drugs used for COVID-19 therapies. Moreover, some ACE inhibitors (ACEi) or angiotensin receptor blockers (ARB), which are used to treat comorbidities in COVID-19 patients, might increase the level of ACE2 in myocardial cells, theoretically, leading to elevated risk for cardiac injury. Therefore, improving the method for patients' treatment should be considered in COVID-19 therapy.

This study has several limitations. First, due to the inclusion and exclusion of a large number of patients, it was inevitable that some important variables were omitted, such as smoking, a history of allergies, and others. Second, we only studied a few patients who exhibited relatively severe illness due to limited medical resources during the epidemic. Third, only patients with clear endpoints were included in the research, some patients who were still in hospital (alive) for treatment were not incorporated into this research, which may result in statistical biases. Last but not the least, about 5% information was lost from the COVID-19 patient list, the missing count variables were supplemented by the median, and the missing categorical variables were supplemented by the mode, which led to biases of the clinical characteristics collection.

## CONCLUSION

In this study, we described the clinical characteristics of COVID-19 patients during hospitalization and innovatively used an RF classification model with these clinical characteristics to predict COVID-19 prognoses. Moreover, we found that LDH concentrations that were higher than 500 U/L, and Myo concentrations higher than 80 ng/ml could be identified as two potential risk factors for mortality of COVID-19 patients. Finally, appropriate treatment should be considered for patients with cardiac and tissue injury.

## ABBREVIATIONS

| | |
|---|---|
| **RF** | Random forest |
| **ROC** | Receiver operating characteristic |
| **AUC** | Area under the ROC curve |
| **IQR** | Interquartile range |
| **ARDS** | Acute respiratory distress syndrome |
| **Lym** | Lymphocyte |
| **Myo** | Myoglobin |
| **NMV** | Noninvasive mechanical ventilation |
| **IMV** | Invasive mechanical ventilation |
| **CRP** | C-reactive protein |

| PCT | Procalcitonin |
|---|---|
| **FIB** | Fibrinogen |
| **BNP** | B-type natriuretic peptide |
| **CK-MB** | Creatine kinase-MB |
| **AST** | Aspartate aminotransferase |
| **ALT** | Alanine aminotransferase |
| **TG** | Triglyceride |
| **LDH** | Lactate dehydrogenase |
| **LMR** | Lymphocyte to monocyte ratio |
| **NLR** | Neutrophil to lymphocyte ratio |
| **Mon** | Monocyte |
| **OOB** | Out-of-bag |
| **Neu** | Neutrophil |

## ACKNOWLEDGEMENTS

We thank the patients and their family members for participating in our study. We appreciate all the efforts of our colleagues in treating the patients.

### Funding

The authors received no funding for this work.

### Competing Interests

The authors declare that they have no competing interests.

### Author Contributions

- Jie Wang conceived and designed the experiments, performed the experiments, analyzed the data, prepared figures and/or tables, authored or reviewed drafts of the paper, and approved the final draft.
- Heping Yu conceived and designed the experiments, performed the experiments, analyzed the data, authored or reviewed drafts of the paper, and approved the final draft.
- Qingquan Hua conceived and designed the experiments, performed the experiments, analyzed the data, prepared figures and/or tables, authored or reviewed drafts of the paper, and approved the final draft.
- Shuili Jing performed the experiments, prepared figures and/or tables, and approved the final draft.
- Zhifen Liu performed the experiments, analyzed the data, authored or reviewed drafts of the paper, and approved the final draft.
- Xiang Peng analyzed the data, prepared figures and/or tables, and approved the final draft.
- Cheng'an Cao conceived and designed the experiments, authored or reviewed drafts of the paper, and approved the final draft.

- Yongwen Luo conceived and designed the experiments, authored or reviewed drafts of the paper, and approved the final draft.

## Human Ethics

The following information was supplied relating to ethical approvals (i.e., approving body and any reference numbers):

Ethics approval was obtained from the Ethics Committee of Wuhan Fourth Hospital (KY 2020-032-01), and informed consent of the study participants was waived by the Ethics Committee of the hospital due to the apparently high transmissibility of the disease.

## Data Availability

Raw data and code are available in the Supplemental Files.

## Supplemental Information

Supplemental information for this article can be found online at http://dx.doi.org/10.7717/peerj.9945#supplemental-information.

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
