# Peer review of "A descriptive study of random forest algorithm for predicting COVID-19 patients outcome"

_PeerJ, doi:10.7717/peerj.9945_

## Round 0.1 · original submission · Major Revisions

Your manuscript has been reviewed and requires several modifications prior to making a decision. The comments of the reviewers are included at the bottom of this letter. Reviewers indicated that the introduction and the methods sections should be improved. The manuscript also needs extensive English editing because there are several typos and grammatical errors. I agree with the evaluation and I would, therefore, request for the manuscript to be revised accordingly. I would also like to suggest the following changes:

-The statistical test (two independent samples t-test and Mann-Whitney-Wilcoxon test) used to check the normality assumption of the data is not correct. Please use the appropriate statistical test to check if the data are normally distributed.

-Change “spss26” to “IBM SPSS Version 26.0”.

-In Line 106 and 112: Correct xiC and pj according to formula.

-In Line 112: Is there a typo in “Pearce correlation”? If yes, please correct the name of the analysis as Pearson’s correlation.

-Please add references to the formulas in Line 103 and in Line 109.

-In Table 3: Correct letters a, b, c, and d in the “statistics” column.

Reviewer 1 ·

Basic reporting

The authors analysed many different clinical measures in 126 patients affected by COVID-19 from the Wuhan Fourth Hospital. They performed a descriptive study by identifying as the main predictors of the clinical outcomes LDH value higher than 500U/L and Myo value higher than 80ng/ml. They tried also to use very different features and a random forest classifier in order to predict the outcome of a specific patient affected by COVID-19 (alive/died). The manuscript is well articulated and written. I really appreciated also the supplementary material for reproducing the results of the paper and the idea to share all the important data for the fights against the virus.

In my opinion figure 1, 2, and 3 can be reviewed. The thickness of the lines is very small and the labels on the axis are not completely clear. The overall impression of the figures is that the resolution should be higher. The impression is the same for the two different equations in the methods section.

In the text there are different typing errors:
Line 112 the authors wrote Pearce correlation: do the authors mean Pearsons correlation?
Line 163 "alvie" instead of, I suppose, alive.
Line 181 the authors wrote, “We divided the data into a training set and a test set at a ratio of 1: 4 (23:96)”. However, by opening the excel file of the training data (S3) there are 96 samples for the training set. In the text, it seems the opposite (23 for training and 96 for testing).

Experimental design

In the experimental design, in my opinion, there are two different points that should be revised:

1) The binary classification task could be affected by the class imbalance problem (16 deaths / 103 no deaths). In this setting, the classifier will be able to identify the patients in the "no deaths" class becoming very specific. In order to overcome this problem, I recommend using an approach based on oversampling procedure (such as ADASYN) [1]-[2] before performing training and testing procedures. It is important to increase the number of observations in the "deaths" class to avoid the problem. Alternatively, it should be possible to reduce the number of observations in the no deaths class until 16. In this way, there are 32 observations. Then, using only two features (LDH and Myo) with a simpler classifier than RF like Linear Discriminant Analysis. This depends on the authors. In the latter case, the classification task becomes feasible. By selecting 90% for the training (28 observations) it is possible to estimate the only 2 weights (one for LDH and one for Myo).

2) Another point is that the results are not presented after a cross-validation procedure. In this way, the results are biased depending on the training and testing set selected. If I try to perform the same analysis with different training and testing sets maybe the results will be different. I recommend performing also a cross-validation procedure. Training and Testing procedure repeated at least 5 times in order to generalize the results for the classification task. The authors should present the mean value of the AUC in the 5 different testing sets. This task is needed independently from the classifier used for the procedure.

References:

[1]-Sciaraffa, Nicolina, et al. "Double-Step Machine Learning-Based Procedure for HFOs Detection and Classification." Brain Sciences 10.4 (2020): 220.

[2]-He, H.; Bai, Y.; Garcia, E.A.; Li, S. ADASYN: Adaptive synthetic sampling approach for imbalanced learning. In Proceedings of the 2008 IEEE International Joint Conference on Neural Networks (IEEE World Congress on Computational Intelligence), Hong Kong, China, 1–8 June 2008; pp. 1322–1328.

Validity of the findings

The results analysis of the data is well explained and the data are the results of months of monitoring during a pandemic. The results could improve for sure the knowledge of the scientific community about the virus.

The statistical analysis presented in this work are strong and robust and gives an exhaustive overview of the effects of COVID-19 on the patients. Furthermore, all the analyses confirmed the fact that it is important to constantly monitor the clinical measures of the patients in order to predict the effects of COVID-19 on the specific patient.

·

Basic reporting

The English language should be improved throughout the manuscript to ensure the readers can understand the text. Some examples of English grammar mistakes include: line 26 (dead or deceased, not died), 27 (was contributed?), unnecessary articles (“the COVID-19”, line 50), lines 59-60, lines 69-71 (run-on sentence), lines 96-97 (incomplete sentence).

Experimental design

Introduction
Lines 63-65: Add further detail on why this is an important topic. What is the benefit of understanding which clinical characteristics predict disease outcome?

Methods:
How were patients selected? What was the study design?
Line 80-81: how was disagreement among the medical record collectors/coders handled? Was each record assessed by one researcher, two or three?
Data collection: what were the outcomes recorded? Discharged vs Dead? Or Alive (including still hospitalized) vs Dead?
Lines 85-88: Were you testing that each continuous variable was normally distributed vs not-normally distributed with t-tests? Or were you comparing the distribution of the continuous variable between the two outcome groups? As stated, it is unclear. Normality should be assessed prior to the application of the t-test. Also, how was the elevated type I error due to multiple comparisons addressed?
Lines 92-98: Please provide additional data on tree fitting, such as pruning and tree growth parameters, number of trees, and the size of the training vs testing sets. Some of this information is in lines 173-189 but should be moved from Results to Methods.

Results (many of these should be addressed by adding detail to Methods section):
Lines 134-135: What were the criteria for determining moderate, severe and critical illness?
Lines 136-137: How many were alive and discharged vs alive and hospitalized? If they were still hospitalized, then the final outcome is unknown.
Lines 140-151: When were laboratory values measured relative to admission? The time of measurement could confound the relationship between laboratory values and outcome. For example, if laboratory values improve during hospitalization for alive patients vs becoming worse for dead patients, then the time of measurement will impact the relationship between laboratory value and outcome.
Lines 196-197: How was the Pearson correlation coefficient incorporated into the trees to avoid over-fitting?
Lines 215-217: How were the 80ng/mL Myo and 500U/L LDH thresholds selected? The Myo threshold is the upper limit of the reference interval (Table 2) but the LDH threshold is much higher than the reference interval. For consistency, I suggest also using the upper limit of the LDH interval or a multiple of the upper limit as the threshold.
Lines 223-225: I don’t think the relationship between poor prognosis and elevated LDH levels is clear. Many patients that survived had levels of LDH outside the reference range (Figure 3B, Table 2).
Line 226: What was the threshold for Myo and LDH associated with these ROC values?
Table 1: Under clinical outcomes there is “Cure Death” and “Death”. Do you mean “Cure” and “Death”? Also, in the text patients are referred to as “Alive” and “Dead”. Which is the actual outcome measured, cure or alive?
Table 2, Headings are unclear: “Patient amount”, “Patient value of deviation”

Validity of the findings

Discussion:
Lines 231-239; 244-253: This seems more appropriate for the Results section
Lines 262-264: This sentence is unclear. What is meant by “high pressure”? High blood pressure?
Line 266-267, 276-277: Where is the partial correlation of the Myo and LDH values with survival? Figure 2 shows partial correlation among the independent variables only.
Please include a discussion of why this work is important and the impact it will have on the treatment of COVID-19 patients or public health response to COVID-19.

Additional comments

This manuscript addresses a timely problem, COVID-19, and could provide insight into patient risk factors for severe disease and death. Random forests are well-suited to this classification problem. However, I have two main concerns with the methodology: (1) censored outcomes, (2) unbalanced data. These concerns need to be addressed prior to publication. Additionally, further detail is required in the Methods section and the English grammar needs to be edited throughout for clarity.


Censoring of outcomes: It is unclear if the classification of alive vs dead represents the final status of each patient. Some patients that were alive on the date of the medical record examination may die at some later point. This can bias the conclusions of the classification algorithm. Therefore, the outcome criteria need to be more precise. Ideally the outcome is discharged alive vs died because this would include only patients that have progressed completely through the clinical disease. Alternatives could be alive vs died after a certain number of days following admission. The time-period following admission should be determined by considering the clinical course of the disease.

Unbalanced data: Algorithms that are trained on unbalanced data (unequal numbers of alive vs dead) can result in poor performance. Since the algorithms minimize the overall error rate, they tend to perform better on the majority class (in this case “alive”) rather than the minority class. In this data set, an algorithm can be correct 87% of the time by just guessing “alive” for every patient. The authors should address this issue. There are many ways to use random forests with unbalanced data, including over-sampling the minority group and under-sampling the majority group.

Reviewer 3 ·

Basic reporting

The authors conducted a retrospective cohort analysis including 126 patients diagnosed with COVID-19 from Wuhan Fourth Hospital. They compared the clinical characteristics of patients between alive and deceased patients. They also conducted a random forest classification model, to find the optimal diagnostic predictors for patients' clinical outcomes between those groups. They identified myo and LDH as the influential features.
1. The English language and grammar should be improved throughout the text.
2. Throughout the manuscript, there are a number of typing errors, such as:
Spaces before and after commas and parentheses;
Line 88 “spss” should be in capital letters;
Line 390 and Table 1: “condnition” must be condition;
Line 394 and Figure 2 caption: “Pearce” Correlation must be Pearson Correlation; etc.
3. The quality of the figures should be improved
4. In the title and conclusion parts, authors are talking about machine learning algorithms for predicting COVID-19 patients outcome, however the only method they use is Random Forest.
5. References are not well-written.

Experimental design

1. The authors should give a clear description and contribution of their workflow, by mentioning their aim in each part.
2. In line 134, they talk about moderate, severe and critical patients. Is there a measure to find the severity?
3. In Figure 2 B, there is no need to show the AUC result for the training data.
4. As shown in Fig 2.3, number of tree is chosen as 500; but they should discuss all other hyperparameters of Random Forest model; and the reasons to choose those parameters in their model.
5. There are some independent variables whose correlations are high (in Figure 2A). Authors should mention whether they conducted an investigation to find the best subset (feature selection).
6. The Random Forest algorithm can be affected by the class imbalance problem. Authors should mention how they handled this problem.
7. In line 226 and Figure 3.C, they should give an information about the threshold for the variables (myo and LDH) associated with the ROC values.

Validity of the findings

1. The authors should mention whether they take the censoring observations into account. (alive/deceased)
2. In lines 91-95 they talk about different R packages. Did they used a different package to build their model; and then another package for validation; it is unclear. For more consistent results, I recommend using a single package, like caret, which can handle all their workflow.
3. They can also improve their results part, talking about the importance of their results, in order to give much deeper information to the readers.

Additional comments

In my opinion, the introduction part should be improved. The authors should talk more about the backgroud of their study, search for other similar studies in the literature; and they should mention about their motivation, and the contribution of this study.
They should address the problems or limitations of their study, such as censoring patients
English grammar needs to be edited throughout the manuscript.

---

## Round 0.2 · Major Revisions

Although reviewer #3 recommended the rejection of current submission, my decision is major revision at this stage. Critically, the reviewer strongly recommended English editing, which I agree with. Besides, there are still a few issues that need to be addressed, I believe. In your rebuttal letter, you mentioned that “after evaluation of the results, the independent variables whose correlations were high didn’t affect the final results of the article.” I am wondering how you made this decision. Please provide detailed information according to the comparisons of the results in the rebuttal letter and manuscript.

Please respond to the reviewer’s comments seriously. I invite you once again to respond to the comments and revise your manuscript.

Reviewer 1 ·

Basic reporting

I would like to thank the authors for the great effort made.

The overall quality of the manuscript has been improved after the review. In particular, the manuscript now is very well articulated and explained. Also, fluency has been improved.
The description of the classification task in the method section is reasonably detailed and all the methodological steps are well explained

Experimental design

The computations have been performed using an oversampling procedure in order to avoid class imbalance problem. The classification results are showed for the training and test sets for avoiding overfitting issues.

Validity of the findings

the use of random forest classifier, with features derived from COVID-19 patients, such as LDH and Myo could allow predicting the prognoses at an early stage and provide scientific data for mortality reduction.

In my opinion, the study of classification accuracy using LDH and Myo as a feature is very important because it provides an objective measure of importance for a clinical measure which can help in identifying the stage of the disease.

Additional comments

Since all the requests made, have been addressed, I recommend this work for the publication.

I found some spelling mistakes that should be checked and I recommend a careful reading of the manuscript.

Reviewer 3 ·

Basic reporting

The authors state they've implemented English editing, however the response letter is poorly written.

Experimental design

In my opinion, the authors try to address an important problem. However, it feels like they still lack necessary skills to conduct the machine learning pipeline for their valuable dataset.

Validity of the findings

The response they provided unfortunately does not satisfy my concerns. I think adding a new author did not improve the presentation of the results.

---

## Round 0.3 · accepted · Accept

The authors addressed the reviewers' concerns and substantially improved the content of MS. So, based on my own assessment as an academic editor, no further revisions are required and the MS can be accepted in its current form.